# PM2.5 Prediction Based on the CEEMDAN Algorithm and a Machine Learning Hybrid Model

**Wenchao Ban** 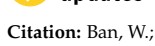 **and Liangduo Shen ***

School of Ocean Engineering Equipment, Zhejiang Ocean University, Zhoushan 316000, China
* Correspondence: slduo@163.com

**Abstract:** The current serious air pollution problem has become a closely investigated topic in people's daily lives. If we want to provide a reasonable basis for haze prevention, then the prediction of PM2.5 concentrations becomes a crucial task. However, it is difficult to complete the task of PM2.5 concentration prediction using a single model; therefore, to address this problem, this paper proposes a fully adaptive noise ensemble empirical modal decomposition (CEEMDAN) algorithm combined with deep learning hybrid models. Firstly, the CEEMDAN algorithm was used to decompose the PM2.5 timeseries data into different modal components. Then long short-term memory (LSTM), a backpropagation (BP) neural network, a differential integrated moving average autoregressive model (ARIMA), and a support vector machine (SVM) were applied to each modal component. Lastly, the best prediction results of each component were superimposed and summed to obtain the final prediction results. The PM2.5 data of Hangzhou in recent years were substituted into the model for testing, which was compared with eight models, namely, LSTM, ARIMA, BP, SVM, CEEMDAN–ARIMA, CEEMDAN–LSTM, CEEMDAN–SVM, and CEEMDAN–BP. The results show that for the coupled CEEMDAN–LSTM–BP–ARIMA model, the prediction ability was better than all the other models, and the timeseries decomposition data of PM2.5 had their own characteristics. The data with different characteristics were predicted separately using appropriate models and the final combined model results obtained were the most satisfactory.

**Keywords:** timeseries data; PM2.5 concentration prediction; CEEMDAN–LSTM–BP–ARIMA coupling model

## 1. Introduction

Atmospheric pollution [1] is closely related to agricultural production and human health. In recent years, with the expansion of industrial production, the problem of atmospheric pollution has become increasingly serious. Thus, we must pay attention to this problem [2]. PM2.5 refers to particles in the air with a particle size less than or equal to 2.5 microns, which can float in the outdoor air for a long time. Its content is a key factor in determining the degree of air pollution. A greater content of PM2.5 in the outdoor air indicates increased pollution. PM2.5 particles are also an important element in the formation of haze, as they are finer particles, are richer in harmful substances, and have a longer transmission distance and life span. Therefore, PM2.5 particles are harmful to human health, the quality of the atmospheric environment, and they have a more direct impact on air quality and visibility. A person's travel and health conditions in haze environments [3] are greatly affected. Therefore, it is necessary to establish accurate, reliable, and effective models to make predictions of atmospheric pollutant concentrations over a long period of time. The prediction results [4–7] can provide guidance for decision-making behavior, in addition to being important for the protection and management of ambient air.

Accurate long-term forecasting of PM2.5 concentration is more important than short-term forecasting, as it allows us more time to deal with the impact of air pollution. At present, the main timeseries forecasting methods are divided into two categories. One is

the traditional probabilistic method [8], which determines the parameters of the model according to theoretical assumptions and prior knowledge of the data. If our actual data and theoretical models do not match, then this method cannot give satisfactory long-term prediction results. There is also the machine learning method [9]. The biggest difference with the traditional probability method is that the machine learning method does not need to determine the model parameters through theoretical assumptions and prior knowledge of the data. The algorithm is used for learning to obtain the law between the model parameters and the data to make predictions. Therefore, deep learning networks have actually surpassed traditional probabilistic prediction methods in many nonlinear modeling fields.

Traditional methods are mostly used in simple applications of the environment, such as threshold autoregressive (TAR) models [10] and hidden Markov models (HMM) [11] since these models are determined on the basis of theoretical assumptions and prior knowledge of the data parameters. Many times we cannot know the previous parameters, resulting in relatively large limitations, which seriously affect the accuracy of prediction.

Machine learning models are generally based on basic algorithms and historical data to build predictive models that can adaptively learn model parameters, obtain laws and relationships between complex data, and conduct simulation training through part of the data to obtain models to predict future development trends. Basic parameter identification algorithms include iterative algorithms [12], particle-based algorithms [13], and recursive algorithms [14].

Machine learning models are divided into shallow networks and deep networks [15,16]. Shallow networks include the short-term prediction method based on the back-propagation (BP) neural network proposed by Ni et al. [17], the improved grey neural network model [18], the radial basis function (RBF) neural network [19], etc. These methods have been used for PM2.5 concentration data, daily average temperature, and other pollutant concentration data.

However, due to the simple structure of the shallow network, it can only achieve short-term prediction performance, and long-term accurate prediction results must be captured using deep neural networks.

Deep machine learning networks show strong learning ability in complex time-series. They have good performance in capturing the high nonlinearity of timeseries data. They can highly abstract data analysis through the multi-nonlinear transformation of complex structures. Timeseries problem-solving functions include long short-term memory (LSTM) [20,21], differential integrated moving average autoregressive model (ARIMA) [22,23], and support vector machine (SVM) [24,25].

Although deep machine learning models have the ability to extract accurate information in complex environments, PM2.5 concentration sequences are datasets with strong randomness and nonlinearity, and the accuracy of long-term prediction needs further development. In recent years, the application of deep learning methods in the field of air pollution has also attracted the attention of researchers, combinatorial approaches to data decomposition have been shown to be effective ways to improve forecasting performance, and various hybrid models have been introduced to forecast nonlinear timeseries data.

For example, Huang and Kuo [26] used a hybrid model based on convolutional neural networks (CNNs) and LSTM to predict PM2.5 concentrations. Rojo used a loess-based seasonal decomposition program [27,28] to decompose the seasonal components of the data for the short-term prediction of air pollen, and Zhang and Li set the wavelet basis function through wavelet decomposition [29,30] to obtain the predicted decomposition information.

The fully adaptive noise ensemble empirical modal decomposition (CEEMDAN) algorithm [31,32] can completely decompose timeseries data into intrinsic mode function (IMF) components with different frequency characteristics [33] and sort them from high frequency to low frequency, which can greatly reduce the complexity of the original data. We believe that the predictive power of a single model is ultimately limited; therefore, we focused on developing a combined model based on data decomposition.

The focus of this study is to improve the long-term prediction accuracy of PM2.5 based on deep learning networks, combined with the CEEMDAN decomposition algorithm, the complexity of the original PM2.5 data is effectively reduced, and to predict all IMF components separately through four different timeseries machine learning models, the BP model, ARIMA model, LSTM model, and SVM model. Different characteristics were used to adapt to different models, and then the optimal prediction results corresponding to each IMF component were superimposed and summed to restore the optimal solution so as to obtain the combination model based on the above models that was most suitable for Hangzhou PM2.5 data. Our innovation priorities were as follows:

(1)　The CEEMDAN decomposition algorithm was introduced for the long-term prediction of PM2.5 timeseries data.
(2)　After the IMF component was obtained using the CEEMDAN decomposition algorithm, an adaptive model was further established according to the characteristics of the IMF component, so as to improve the accuracy of long-term prediction.
(3)　This method of further predictive analysis of the IMF components can become a new framework, which can be applied for the prediction of more similar data, thus obtaining accurate long-term prediction results.

## 2. Materials and Methods

The air quality data used in this paper came from the air quality historical data query website (https://www.aqistudy.cn/historydata/, accessed on 19 July 2022), and the daily average PM2.5 concentration data records from December 2013 to December 2021 were selected from Hangzhou, Zhejiang Province, and Kunming, Yunnan Province. The samples were both 2952. The sampling interval was 1 day (the daily average data for each day are obtained from the arithmetic mean of the 24-h effective hourly concentration data of all single stations in the city where they are located). The PM2.5 unit was ug/m$^3$. The first 90% of the sample data was selected for simulation training, and the last 10% of the sample data was used to test the prediction results; i.e., we used the PM2.5 concentration data for 2952 days (December 2013–February 2021) to predict the trend of PM2.5 concentration data in the next 295 days (March 2021–December 2021). Lastly, in the 295-day forecast results, we intercepted the 50-day model forecast results with more obvious comparisons for graphical display and comparative analysis. The timeseries data of PM2.5 concentration in Hangzhou are shown in Figure 1, which shows that the PM2.5 concentration in Hangzhou had obvious changes and fluctuations.

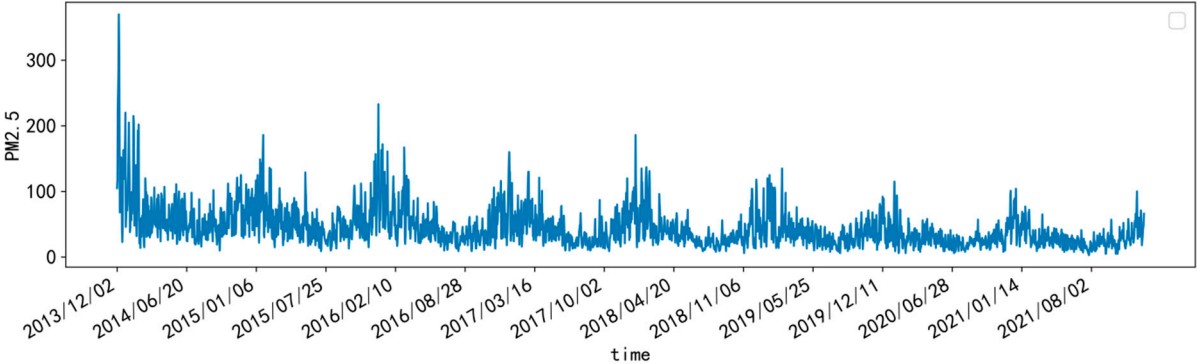

**Figure 1.** Hangzhou PM2.5 data sequence diagram.

### 2.1. Introduction to the Model

2.1.1. Fully Adaptive Noise Ensemble Empirical Modal Decomposition (CEEMDAN) Algorithm

The CEEMDAN algorithm is mostly used in the field of forecasting. It can completely decompose the original data, with strong volatility, into several intrinsic mode function (IMF) components with different frequency characteristics, thereby reducing the volatility of

the data and improving the prediction accuracy. The CEEMDAN algorithm has three major advantages. The first is its completeness. In other words, the original data can be obtained by adding and summing the components decomposed by the algorithm, which is beyond the reach of many decomposition algorithms. Secondly, the CEEMDAN algorithm has a faster calculation speed, which can effectively improve the operation speed of the program. Lastly, the CEEMDAN algorithm has a better modal decomposition effect, preventing the occurrence of multiple low-frequency components with small amplitudes, which are of little significance for data analysis.

Let $E_i(\cdot)$ be the $i$th eigenmode component obtained through empirical mode decomposition (EMD). Then, the $i$th eigenmodal component obtained through CEEMDAN decomposition is $\overline{C_i(t)}$. $V^j$ denotes the Gaussian white noise signal conforming to the standard normal distribution. $j = 1, 2, \ldots, N$ is the frequency at which white noise is added, $\varepsilon$ is the standard table of white noise and $y(t)$ is the data to be decomposed. The CEEMDAN decomposition steps are described below.

(1)  The new signal $y(t) + (-1)^q \varepsilon^j(t)$ is obtained by adding Gaussian white noise to the signal to be decomposed $y(t)$, where $q$ = 1.2. The EMD decomposition of the new signal yields the first order eigenmode components.

$$E(y(t) + (-1)^q \varepsilon v^j(t)) = C_1^j(t) + r^j \tag{1}$$

(2)  The first eigenmodal component of the CEEMDAN decomposition is derived by taking the resulting $N$th modal component and balancing it overall.

$$\overline{C_1(t)} = \frac{1}{N} \sum_{j=1}^{N} C_1^j(t) \tag{2}$$

(3)  The residuals are calculated after subtracting the first modal component.

$$r_1(t) = y(t) - \overline{C_i(t)} \tag{3}$$

(4)  By adding a new signal of positive and negative paired Gaussian white noise to $r_1(t)$ and using the new signal as a medium to start the EMD decomposition, the first-order modal component, $D_1$, can be derived, which generates the second eigenmodal component of the CEEMDAN decomposition as follows:

$$\overline{C_2(t)} = \frac{1}{N} \sum_{j=1}^{N} D_1^j(t). \tag{4}$$

(5)  The residuals are calculated after subtracting the second modal component from the results.

$$r_2(t) = r_1(t) - \overline{C_2(t)} \tag{5}$$

(6)  The above process is repeated until the obtained residual signal is always a monotonic function and cannot be further decomposed. Then, the calculation is completed. The number of eigenmodal components obtained at this point is $K$. The original signal is then decomposed as

$$y(t) = \sum_{K=1}^{K} \overline{C_K(t)} + r_K(t). \tag{6}$$

An exploded view of the CEEMDAN algorithm is shown in Figure 2. The original PM2.5 concentration timeseries data in Hangzhou were completely decomposed into nine intrinsic mode function (IMF) components and residuals and then displayed in order from top to bottom according to high and low frequency. Then, the difference between each IMF component and residual was substituted into the machine learning timeseries model for

prediction. Lastly, the final prediction result was obtained by superimposing and summing the predicted IMF components. The flowchart is shown in Figure 3.

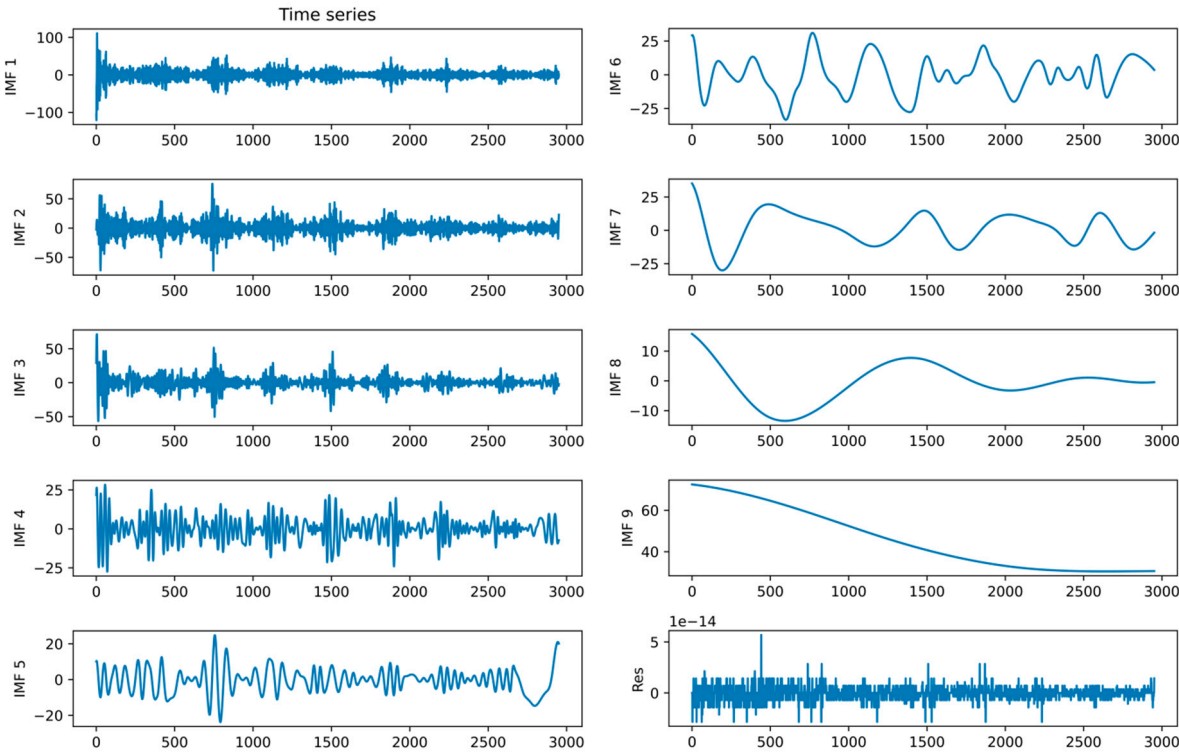

**Figure 2.** Splitting result diagram based on a fully adaptive noise ensemble empirical modal decomposition (CEEMDAN) algorithm.

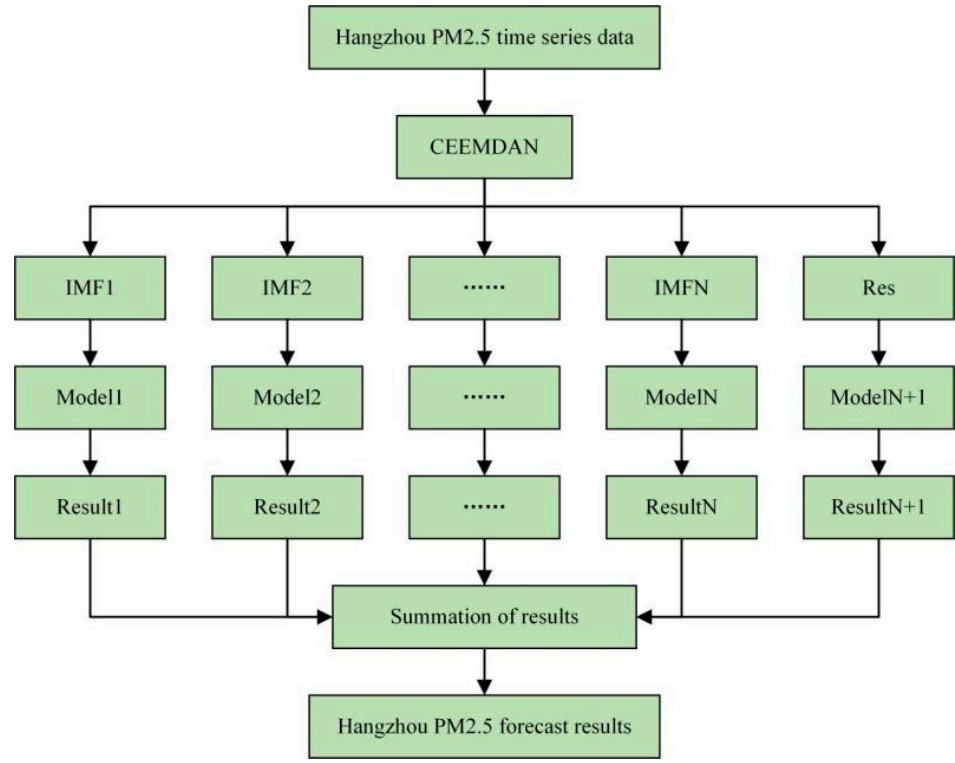

**Figure 3.** CEEMDAN model flowchart.

### 2.1.2. Long Short-Term Memory (LSTM) Model

LSTM has a strong ability to remember data in a delayed manner; when it grasps the current content, it acquires the relationships existing in the data over a longer time span to achieve long-term memory, enabling more accurate predictions of their development. An LSTM cell has three gates, called the forget gate, the input gate, the and output gate. Figure 4 shows the cell body of the neuronal structure.

$$f_t = \sigma(W_f * [h_{t-1}, x_t] + b_f) \tag{7}$$

Here, $f_t$ is the forget gate, $\sigma$ is the standard sigmoid activation function, $h_{t-1}$ indicates the unit value at the previous time, $W_f$ is the weight value, $x_t$ is the input value at time $t$, and $b_f$ is the vector of bias value.

$$i_t = \sigma(W_i * [h_{t-1}, x_t] + b_i) \tag{8}$$

$$\widetilde{C}_t = \tanh(W_c * [h_{t-1}, x_t] + b_c) \tag{9}$$

Here, $i_t$ is the input gate, $\widetilde{C}_t$ is the current input state unit, and $\tanh(\cdot)$ is the hyperbolic tangent activation function.

$$C_t = f_t * C_{t-1} + i_t * \widetilde{C}_t \tag{10}$$

Here, $C_t$ is the state cell of the hidden layer at moment $t$, $f_t$ is the degree of the last forget function about information $C_{t-1}$, $i_t$ is the degree to which $\widetilde{C}_t$ is to be added, and $C_t$ is the state of the cell obtained.

$$O_t = \sigma(W_0[h_{t-1}, x_t] + b_0) \tag{11}$$

$$h_t = o_t * \tanh(C_t) \tag{12}$$

Here, $O_t$ is the current output gate.

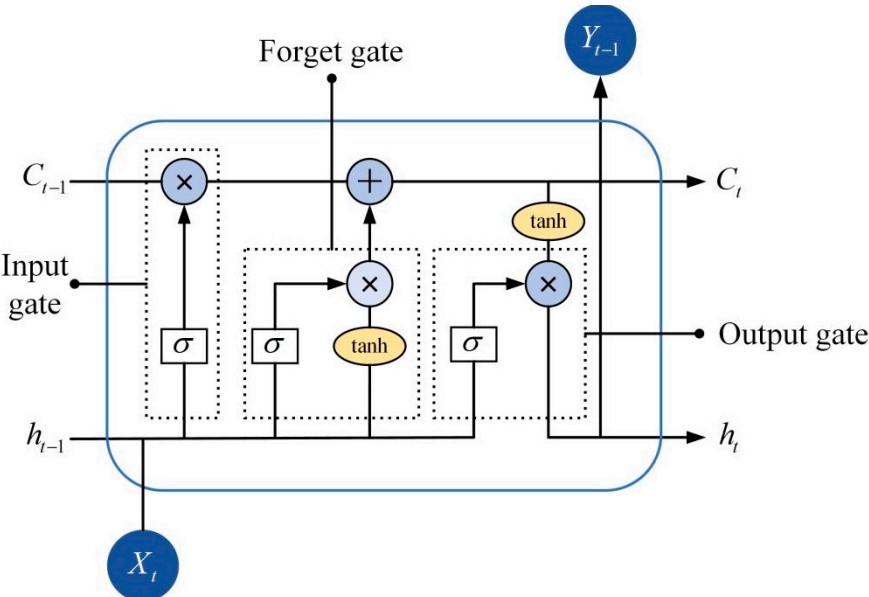

**Figure 4.** Long short-term memory (LSTM) neuronal structure.

### 2.1.3. Differential Integrated Moving Average Autoregressive Model (ARIMA)

The ARIMA model, known as the differential integrated moving average autoregressive model, is the most commonly used demand forecasting model in all industries. It treats the original timeseries data to be forecasted as a random dataset of equal length, uses mathematical methods to identify the characteristics of this dataset, and describes them by means of a numerical model. After modeling, it is possible to predict future values from known values. The flowchart of the ARIMA model is shown in Figure 5.

Modeling flowchart of ARIMA:

**Figure 5.** Differential integrated moving average autoregressive model (ARIMA) flowchart.

The ARIMA (p, d, q) model yields the following expression:

$$\left(1 - \sum_{i=1}^{p} \phi_i L^i\right)(1 - L^d)X_t = \left(1 + \sum_{i=1}^{q} \theta_i L^i\right)\varepsilon_t, \tag{13}$$

where L is the lag operator, and p, d, and q represent the three basic parameters in the model; p is the number of lags of the original data itself in the ARIMA model, d is a positive integer parameter describing the number of orders of differentiation needed to stabilize the original data, and q is the number of lags of the prediction error used in the prediction model.

### 2.1.4. Backpropagation (BP) Neural Network Model

BP neural networks are a class of multilayer feedforward neural networks trained by error backpropagation, and they are one of the most popular neural network architectures in use today. Their network structure can be divided into an input layer, an implicit layer, and an output layer. The essence of this model is to convert the sample data from an input–output problem into a nonlinear optimization problem and use certain methods to change the weights along the negative trend of the error function. Figure 6 shows the structure of the BP neural network model.

Its mathematical expression is as follows:

$$
\begin{aligned}
y = \ & w_{11}^{(2,3)} * \tan sig(w_{11}^{(1,2)} * x_1 + w_{21}^{(1,2)} * x_2 + b_1^{(2)}) \\
& + w_{21}^{(2,3)} * \tan sig(w_{12}^{(1,2)} * x_1 + w_{22}^{(1,2)} * x_2 + b_2^{(2)}) \\
& + w_{31}^{(2,3)} * \tan sig(w_{13}^{(1,2)} * x_1 + w_{23}^{(1,2)} * x_2 + b_3^{(2)}) + b_1^{(3)}
\end{aligned}
\tag{14}
$$

where $w$ is the weight value, $b$ is the threshold value, $w_{11}^{(2,3)}$ represents the weight from the first node of the second layer to the first node of the third layer, and $b_1^{(2)}$ represents the threshold of the first node of the second layer.

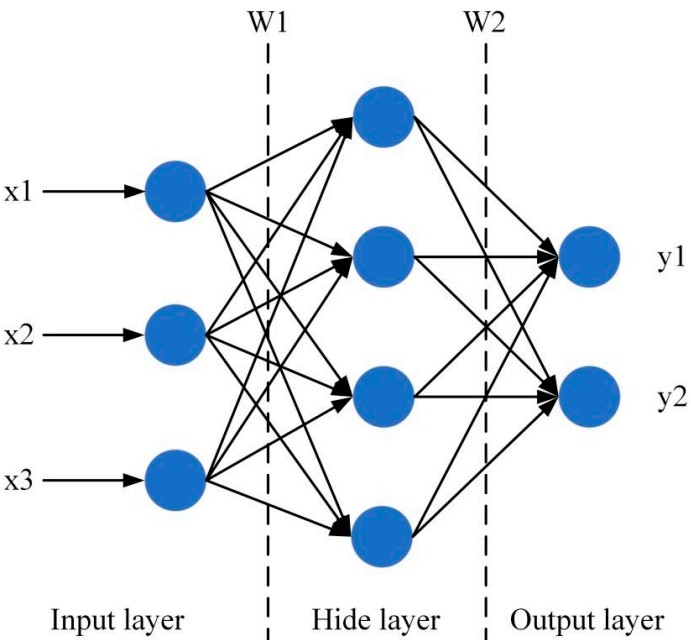

**Figure 6.** Backpropagation (BP) neural network structure diagram.

2.1.5. Support Vector Machine (SVM) Model

The SVM is essentially a small sample classification method. It maximizes the spatial distance of the samples by finding an optimal partitioning "plane", and it is superior in solving nonlinear high-dimensional spatial problems. It is often used in pattern recognition, function approximation, etc. The principle of the SVM method is shown in Figure 7.

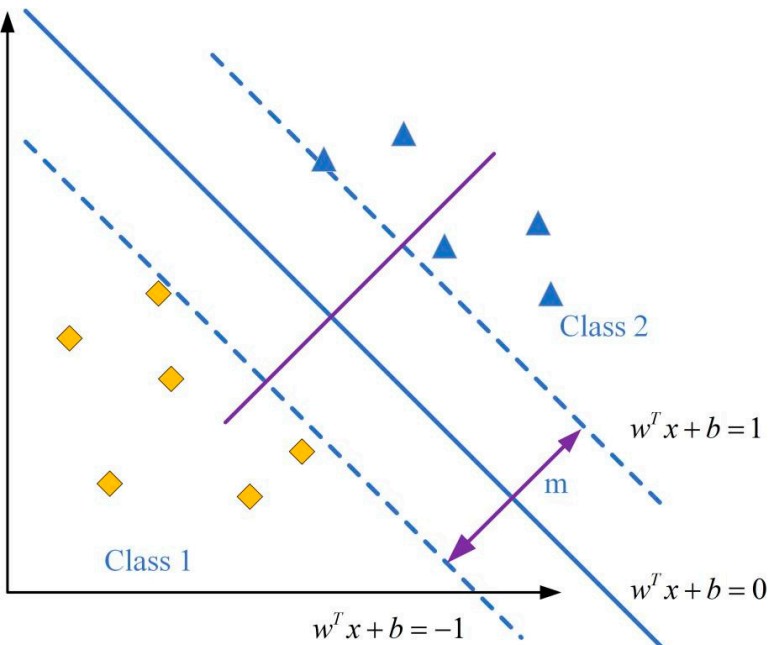

**Figure 7.** Support vector machine (SVM) schematic.

The expression of the optimal partition 'plane' is as follows:

$$w^T x + b = 0. \tag{15}$$

The two plane expressions that are parallel and equal to the optimal dividing 'plane' are expressed as follows:

$$w^T x + b = 1, \tag{16}$$

$$w^T x + b = -1. \tag{17}$$

### 2.1.6. Reference for Setting Important Parameters of the Model

In this model, the input dataset was the original PM2.5 timeseries data, the training set is the first 90% of the original data, and a select 10% of the training set data for 10-fold cross testing, reduce the chance caused by a random division, improve its generalization ability, and improve the efficiency of data use. the LSTM model was implemented using the keras package in Python 3.6. We set the most important parameter reference values to include a hidden layer with 300 dimensions, the 'relu' activation function, look back of 10, 'Adam' optimizer, batch size of 64, and 300 epochs. For the ARIMA model, we chose to use the Akaike information criterion to formulate the reasonable order of p and q, only limiting the parameter p. The maximum value of q was 3, with a scrolling step of 30. The model automatically determined the best order within three operations. The reference order of the BP neural network model function prediction order was 30, and the hidden layer reference was three layers. In the model prediction of SVM, a scrolling step of 30, the KernelFunction was set to 'polynomial', the KernelScale was set to 10, and the BoxConstraint was set to 5.

### 2.1.7. Model Evaluation Indices

In order to effectively compare the parameters of the models, we chose the root-mean-square error (RMSE), the mean absolute error (MAE), and the goodness of fit (*R*-squared, coefficient of determination) to judge the prediction accuracy of the models. The units of RMSE and MSE were the same as that of the PM2.5 concentration data, i.e., ug/m$^3$. Smaller values indicate a greater measurement accuracy. A closer value of $R^2$ to 1 indicates a better fit. The expressions are shown in Equations (18)–(20):

$$\text{RMSE} = \sqrt{\frac{1}{m}\sum_{i=1}^{m}(y_i - \hat{y}_i)^2}, \tag{18}$$

$$\text{MAE} = \frac{1}{m}\sum_{i=1}^{m}\left|y_i - \hat{y}_i\right|, \tag{19}$$

$$R^2 = 1 - \frac{\sum_{i=1}^{m}(y_i - \hat{y}_i)^2}{\sum_{i=1}^{m}(y_i - \overline{y})^2}, \tag{20}$$

where $m$ is the number of test samples, and $y_i$ and $\hat{y}_i$ are the original and predicted values of the data, respectively.

## 3. Results and Analysis

In this study, we first compared the prediction effectiveness of individual models for PM2.5.

The PM2.5 prediction results of the four single models are shown in Figure 8, and the error results are shown in Table 1. It can be concluded that the basic trend of the future PM2.5 concentration could be predicted simply by using single models, but there were clearly large errors, especially at the peaks and troughs. The ARIMA model was the best model for predicting PM2.5 in Hangzhou among the four single models, with an RMSE of 10.09, an MAE of 7.51, and an $R^2$ of 0.46.

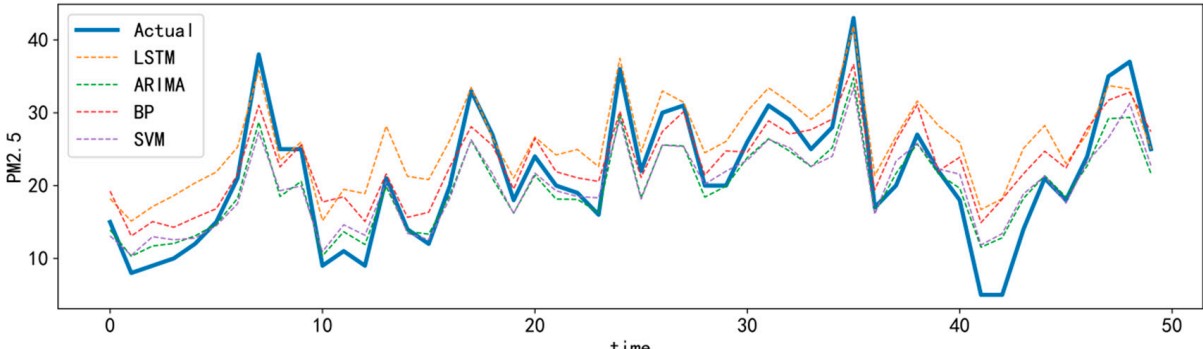

**Figure 8.** Comparison of single model prediction results.

**Table 1.** Single model error comparison.

|       | LSTM  | ARIMA | BP    | SVM   |
|-------|-------|-------|-------|-------|
| RMSE  | 10.27 | 10.09 | 10.55 | 10.61 |
| MAE   | 7.88  | 7.51  | 8.21  | 7.93  |
| $R^2$ | 0.42  | 0.46  | 0.37  | 0.43  |

We then optimized the single models using the CEEMDAN algorithm due to its predictive power. Figure 9 and Table 2 show the prediction and error comparison results for the CEEMDAN–LSTM, CEEMDAN–ARIMA, CEEMDAN–BP, and CEEMDAN–SVM models.

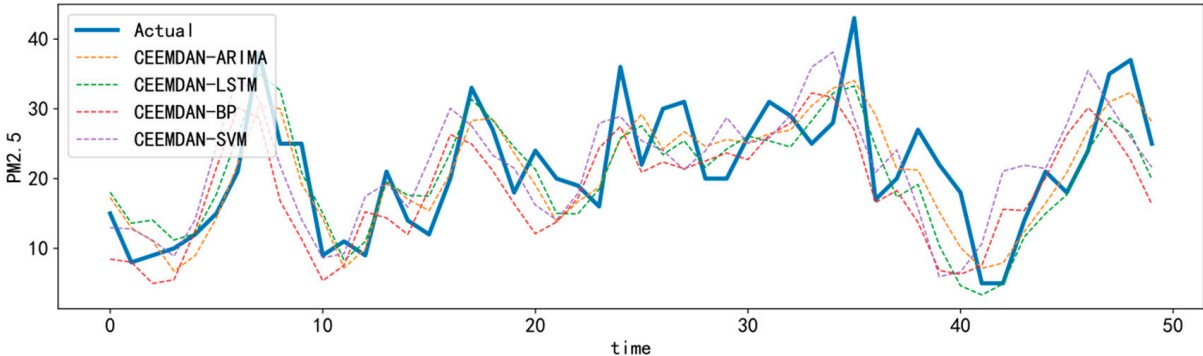

**Figure 9.** Comparison of prediction results using coupled models based on the CEEMDAN algorithm.

**Table 2.** Comparison of coupled model errors based on the CEEMDAN algorithm.

|       | CEEMDAN–LSTM | CEEMDAN–ARIMA | CEEMDAN–BP | CEEMDAN–SVM |
|-------|--------------|---------------|------------|-------------|
| RMSE  | 7.28         | 6.27          | 6.90       | 6.27        |
| MAE   | 5.71         | 4.95          | 5.48       | 4.88        |
| $R^2$ | 0.70         | 0.74          | 0.67       | 0.77        |

It can be concluded that the prediction ability of the coupled models processed using the CEEMDAN algorithm was greatly improved, and the trend of PM2.5 concentration in the future could be better predicted. Every error metric in the coupled model was better than that of the single models in Table 1. The fitting result of the CEEMDAN–SVM model was best, with an RMSE value of 6.27, an MAE value of 4.88, and an $R^2$ of 0.77. The comparison between the coupled models, combined with the CEEMDAN algorithm, and the simple models without the CEEMDAN algorithm fully reflect the superior performance of the CEEMDAN algorithm in improving the prediction accuracy of PM2.5.

Without the CEEMDAN algorithm, the predictive ability of the ARIMA model was better than that of the SVM model; whereas, with the CEEMDAN algorithm, the prediction ability of the CEEMDAN–SVM model was better than that of the CEEMDAN–ARIMA model. Their RMSE values were the same, but the MAE values of the CEEMDAN–SVM model were better. Therefore, we chose to compare the prediction results of each modal component after decomposition using the CEEMDAN algorithm in detail.

We used the CEEMDAN algorithm to completely decompose the original PM2.5 concentration data in Hangzhou into nine IMF components and residuals according to their frequency from high to low, and then substituted each IMF component and residual into the LSTM, ARIMA, SVM, and BP neural network models for prediction. The final prediction result was obtained by stacking and summing. The comparison results are shown in Tables 3–5. The LSTM network performed best in IMF1 component prediction. The BP neural network performed best in predicting the IMF2 components. The prediction performance of the ARIMA model was excellent for the IMF3–9 components. Since the residual value was minuscule, the value of all models when predicting the residual was infinitely close to 0. It can be concluded that the model corresponding to each data type with the best performance was different, and no model could be used for all components. Therefore, data types with different frequencies should be matched with different prediction models, and the optimal solution for each IMF component prediction was obtained by constructing the CEEMDAN–LSTM–BP–ARIMA model. The prediction results obtained from the experiment are shown in Figure 10, and the error results are shown in Table 6. The fitting degree in the figure is high, and the various error indicators are significantly better than all the previous models.

**Table 3.** Comparison of RMSE values of split result graphs based on the CEEMDAN algorithm.

|  | LSTM (RMSE) | ARIMA (RMSE) | SVM (RMSE) | BP (RMSE) |
|---|---|---|---|---|
| IMF1 | 5.84 | 6.16 | 6.12 | 6.90 |
| IMF2 | 1.76 | 1.53 | 1.80 | 1.52 |
| IMF3 | 0.27 | 0.25 | 0.71 | 0.29 |
| IMF4 | 0.15 | 0.04 | 0.43 | 0.05 |
| IMF5 | 0.10 | $1.57 \times 10^{-3}$ | 0.20 | $1.9 \times 10^{-3}$ |
| IMF6 | 0.04 | $6.66 \times 10^{-5}$ | 0.46 | $1.1 \times 10^{-3}$ |
| IMF7 | 0.06 | $3.72 \times 10^{-6}$ | 0.77 | $1.2 \times 10^{-3}$ |
| IMF8 | 0.01 | $3.81 \times 10^{-8}$ | 0.14 | $3.21 \times 10^{-4}$ |
| IMF9 | 0.13 | $1.38 \times 10^{-8}$ | 1.34 | 0.09 |
| Residual | 0.00 | 0 | 0 | 0 |

**Table 4.** Comparison of MAE values of split result graphs based on the CEEMDAN algorithm.

|  | LSTM (MAE) | ARIMA (MAE) | SVM (MAE) | BP (MAE) |
|---|---|---|---|---|
| IMF1 | 4.70 | 4.97 | 4.75 | 5.51 |
| IMF2 | 1.37 | 1.20 | 1.31 | 1.17 |
| IMF3 | 0.19 | 0.17 | 0.53 | 0.21 |
| IMF4 | 0.13 | 0.02 | 0.35 | 0.03 |
| IMF5 | 0.05 | $1 \times 10^{-3}$ | 0.14 | $1 \times 10^{-3}$ |
| IMF6 | 0.03 | $2.17 \times 10^{-5}$ | 0.38 | $7.83 \times 10^{-4}$ |
| IMF7 | 0.05 | $7.36 \times 10^{-7}$ | 0.68 | $8.78 \times 10^{-4}$ |
| IMF8 | 0.01 | $7.88 \times 10^{-9}$ | 0.10 | $3.10 \times 10^{-4}$ |
| IMF9 | 0.09 | $9.35 \times 10^{-9}$ | 1.34 | 0.074 |
| Residual | 0.00 | 0.00 | 0.00 | 0.00 |

**Table 5.** Comparison of $R^2$ values of split result graphs based on CEEMDAN algorithm.

|  | LSTM ($R^2$) | ARIMA ($R^2$) | SVM ($R^2$) | BP ($R^2$) |
|---|---|---|---|---|
| IMF1 | 0.20 | 0.16 | 0.17 | 0 |
| IMF2 | 0.88 | 0.91 | 0.88 | 0.91 |
| IMF3 | 0.99 | 1.00 | 0.98 | 1.00 |
| IMF4 | 1.00 | 1.00 | 0.99 | 1.00 |
| IMF5 | 1.00 | 1.00 | 1.00 | 1.00 |
| IMF6 | 1.00 | 1.00 | 1.00 | 1.00 |
| IMF7 | 1.00 | 1.00 | 1.00 | 1.00 |
| IMF8 | 1.00 | 1.00 | 1.00 | 1.00 |
| IMF9 | 1.00 | 1.00 | 1.00 | 1.00 |
| Residual | 0.00 | 0.00 | 0.00 | 0.00 |

**Table 6.** CEEMDAN–LSTM–BP–ARIMA model error comparison (Hangzhou).

|  | CEEMDAN–LSTM–BP–ARIMA |
|---|---|
| RMSE | 5.9 |
| MAE | 4.63 |
| $R^2$ | 0.79 |

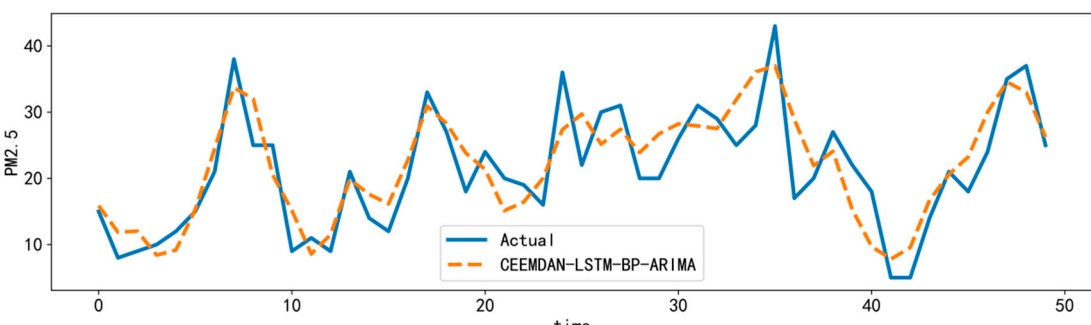

**Figure 10.** CEEMDAN–LSTM–BP–ARIMA prediction comparison (Hangzhou).

In order to better reflect the applicability of the model, and to prevent the occurrence of chance, we substitute the data from Kunming, Yunnan Province, into the optimal model for validation. The prediction results are shown in Figure 11, and the error results are shown in Table 7. We can obviously conclude that the optimal model applicable to Hangzhou PM2.5 prediction is also applicable to Kunming, and the prediction results fit well with the actual values. Therefore, in this experiment, it can be said that the model combining the machine learning time series model with the CEEMDAN algorithm is the ideal model for predicting PM2.5 concentration.

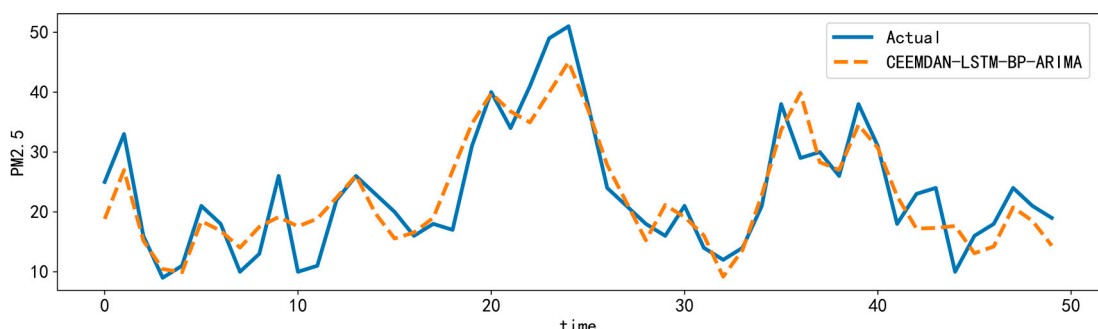

**Figure 11.** CEEMDAN–LSTM–BP–ARIMA prediction comparison (Kunming).

**Table 7.** CEEMDAN–LSTM–BP–ARIMA model error comparison (Kunming).

|  | CEEMDAN–LSTM–BP–ARIMA |
| --- | --- |
| RMSE | 4.55 |
| MAE | 3.66 |
| $R^2$ | 0.79 |

Comparing various methods, Qian [8] et al. used the traditional probabilistic method for PM2.5 concentration prediction, which was able to provide highly time-resolved particle concentrations, but required high parameters of the individual data itself, and required the combination of meteorological variables, land use terms, and spatial and temporal lag terms. In contrast, the learning ability of simple time series models such as LSTM, ARIMA, SVM, and BP is limited, and there is an upper limit to their ability to handle anomalous data. Zhao [20] et al. used LSTM models to model the local variation of PM2.5, Chen [18] et al. used BP neural networks for a 3-h short-term prediction of PM2.5 concentrations, and the experimental results proved that the single machine learning time series model has certain PM2.5 concentration prediction ability, which is mainly superior in the short term or locally, and the long-term results are not satisfactory. The CEEMDAN algorithm has been well applied in the hands of Rongbin [32] et al. The algorithm has good decomposition integrity by decomposing the original signal with complexity and nonsmoothness into eigenmodal components (IMF), and the decomposition completes with significantly lower data values, so the decomposed training data can better improve the prediction accuracy when applied to a single neural network model.

**4. Conclusions**

In recent years, air quality problems have had a serious impact on people's normal life. Environmental problems such as PM2.5 have received more and more attention and PM2.5 is characterized by a strong multilateral and a strong randomness. Thus, accurate long-term PM2.5 concentration prediction remains a formidable challenge for us.

In this study, we proposed a way to combine the CEEMDAN algorithm with the LSTM model, ARIMA model, BP neural network, and SVM model to predict the PM2.5 concentration. The results of various evaluation indicators showed that all models based on the CEEMDAN algorithm improved the prediction accuracy to varying degrees compared with the original simple models. The introduction of the CEEMDAN algorithm can provide new inspiration for a PM2.5 prediction, and the CEEMDAN algorithm can perhaps be combined with additional timeseries machine learning models. In this experiment, the predictive performance of the coupled model was higher than that of the single model.

Secondly, we discovered a new application of the IMF components obtained using the CEEMDAN algorithm. We carried out LSTM, ARIMA, BP neural network, and SVM modeling and prediction for each IMF component of the PM2.5 concentration timeseries data. The optimal prediction results of the components were added and summed. In this paper, the CEEMDAN-LSTM-BP-ARIMA model obtained the ideal results for a PM2.5 concentration prediction in Hangzhou and Kunming. Compared with the other models, the long-term prediction accuracy was significantly improved. Applying a single model is not optimal. We found that the best models differed according to IMF components, whereby a combination of timeseries machine learning models obtained the best prediction accuracy. We believe that this method also has good generalizability and can be used to predict additional characteristics such as wind speed and other pollutant concentrations.

**Author Contributions:** Conceptualization, W.B.; methodology, W.B.; software, W.B.; validation, W.B.; formal analysis, W.B.; investigation, W.B.; resources, W.B.; data curation, W.B.; writing—original draft preparation, W.B.; writing—review and editing, W.B.; visualization, W.B.; supervision, L.S.; project administration, L.S.; funding acquisition, L.S. All authors have read and agreed to the published version of the manuscript.

**Funding:** This research was financially supported by Provincial scientific research fund for basic research (2021JZ008) and General Projects of Zhoushan Science and Technology.

**Institutional Review Board Statement:** Not applicable.

**Informed Consent Statement:** Not applicable.

**Data Availability Statement:** Not applicable.

**Conflicts of Interest:** The authors declare no conflict of interest.

## Abbreviations

| Abbreviation | Full Name |
|---|---|
| CEEMDAN | Complete EEMD with adaptive noise |
| LSTM | Long short-term memory |
| BP | Backpropagation |
| ARIMA | Differential integrated moving average autoregressive |
| SVM | Support vector machine |
| IMF | Intrinsic mode function |

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
