# Peer review of "PM2.5 Prediction Based on the CEEMDAN Algorithm and a Machine Learning Hybrid Model"

_sustainability, doi:10.3390/su142316128_

Round 1
Reviewer 1 Report
The current manuscript predicted the PM2.5 based on a combination of the selected algorithms in Hangzhou. But this manuscript is more like a review, or an experiment report; and there are lots of work needed to be done for this manuscript, if authors want to publish it. The reasons are:
1. The manuscript didn’t detail the input datasets; for example, I, as a reviewer, don’t see the introduction of the independent variable, which is questionable.
2. In a 12-pages manuscript, the results only cover 3 pages from Page 8 to 10. There is no any analysis and conclusion based on the experiment results, for example, the different features of the different original algorithms, and the relation between the selected algorithms and the run-time mechanism of PM2.5 model in Hangzhou.
3. After reviewing this manuscript, we still don’t know the reason why authors choose this combination of the algorithms, when there is only an exhibition of the experiment accuracy. Because the good accuracy of your method must be explainable and adaptable, which need to be proved by lots of experiment and analysis.
This paper still has major issues that need to be addressed, which includes the designation of the algorithm, the introduction of the experimental data, and the addition of the analysis of the experiment results. The aforementioned suggestions are just a part. Considering so much work for improving the manuscript. I don’t recommend the publication of this paper on the journal. And, resubmission may be an option after the improvement for this manuscript.
Author Response
Dear reviewer:
Thank you for your decision and constructive comments on my manuscript entitled“PM2.5 Prediction Based on CEEMDAN Algorithm and Deep Learning Hybrid Model”(sustainability-1850342).Those comments are all valuable and very helpful for revising and improving our paper, as well as the important guiding significance to our researches. We have studied comments carefully and have made corrections which we hope meet with approval. Revised portions are marked in red on the paper.The corrections in the paper and the responses to the reviewer’s comments are as flowing.
- The manuscript didn’t detail the input datasets; for example, I, as a reviewer, don’t see the introduction of the independent variable, which is questionable.
Response:
Thank you for this comment.I'm sorry that we didn't clearly explain the input data set, causing you to have doubts about it. In fact, the independent variable of the input data set is itself. The machine learning models used in our article are all time series prediction models, and the input is arranged in time order. The time series prediction model can automatically extract the time series features of PM2.5 sample data and make predictions,and does not use the method that requires additional input variables. We will rewrite the introductory part (page 1 line 30 -page 3 line 19) of the paper to make the relationship between our model and the data clearer for readers, and add a description of the input dataset to the Materials and methods part of the paper to complete the paper (page 3, line 21 - 43).
Modifications in the original text are as follows (page 1 line 30 -page 3 line 19):
Atmospheric pollution [1] is closely related to agricultural production and human health. In recent years, with the expansion of industrial production, the problem of atmospheric pollution has become increasingly serious; thus, we must pay attention to this problem [2]. PM2.5 refers to particles in the air with a particle size less than or equal to 2.5 microns, which can float in the outdoor air for a long time. Its content is a key factor in determining the degree of air pollution; a greater content of PM2.5 in the outdoor air indicates increased pollution. PM2.5 particles are also an important element in the formation of haze, as they are finer particles, are richer in harmful substances, and have a longer transmission distance and life span; therefore, they are harmful to human health and the quality of the atmospheric environment, and they have a more direct impact on air quality and visibility. People’s travel and health conditions in haze environments [3] are greatly affected. Therefore, it is necessary to establish accurate, reliable, and effective models to make predictions of atmospheric pollutant concentrations over a long period of time. The prediction results [4–7] can provide guidance for decision-making behavior, in addition to being important for the protection and management of ambient air.
Accurate long-term forecasting of PM2.5 concentration is more important than short-term forecasting, as it allows us more time to deal with the impact of air pollution. At present, the main timeseries forecasting methods are divided into two categories. One is the traditional probabilistic method [8], which determines the parameters of the model according to theoretical assumptions and prior knowledge of the data. If our actual data and theoretical models do not match, then this method cannot give satisfactory long-term prediction results. There is also the machine learning method [9]. The biggest difference with the traditional probability method is that it does not need to determine the model parameters through theoretical assumptions and prior knowledge of the data. The algorithm is used for learning to obtain the law between model parameters and data to make predictions. Therefore, deep learning networks have actually surpassed traditional probabilistic prediction methods in many nonlinear modeling fields.
Traditional methods are mostly used in simple applications of the environment, such as threshold autoregressive (TAR) models [10] and hidden Markov models (HMM) [11], since these models are determined on the basis of theoretical assumptions and prior knowledge of the data parameters, and many times we cannot know the previous parameters, resulting in relatively large limitations, which seriously affect the accuracy of prediction.
Machine learning models are generally based on basic algorithms and historical data to build predictive models that can adaptively learn model parameters, obtain laws and relationships between complex data, and conduct simulation training through part of the data to obtain models to predict future development trends. Basic parameter identification algorithms include iterative algorithms [12], particle-based algorithms [13] and recursive algorithms [14].
Machine learning models are divided into shallow networks and deep networks [15,16]. Shallow networks are short-term prediction methods based on the backpropagation (BP) neural network proposed by Ni et al. [17], which has been used for PM2.5 concentration data, daily average temperature, and other pollutant concentration data.
However, due to the simple structure of the shallow network, it can only achieve short-term prediction performance, and long-term accurate prediction results must be captured using deep neural networks.
Deep machine learning networks show strong learning ability in complex timeseries. They have good performance in capturing the high nonlinearity of timeseries data. They can highly abstract data analysis through multi-nonlinear transformation of complex structures. Timeseries problem-solving functions include long short-term memory (LSTM) [18,19], differential integrated moving average autoregressive model (ARIMA) [20,21], and support vector machine (SVM) [22,23].
Although deep machine learning models have the ability to extract accurate information in complex environments, PM2.5 concentration sequences are datasets with strong randomness and nonlinearity, and the accuracy of long-term prediction needs further development. In recent years, combinatorial approaches to data decomposition have been shown to be effective ways to improve forecasting performance, and various hybrid models have been introduced to forecast nonlinear timeseries data.
For example, the seasonal decomposition program based on loess [24,25] can perform short-term prediction of air pollen according to the decomposed seasonal components, and wavelet decomposition [26, 27] can obtain decomposition information by setting the wavelet basis function for prediction.
Fully adaptive noise ensemble empirical modal decomposition (CEEMDAN) algorithm[28,29] can decompose timeseries data into intrinsic mode function (IMF) components with different frequency characteristics [30], and sort them from high frequency to low frequency, which can greatly reduce the complexity of the original data. We believe that the predictive power of a single model is ultimately limited; therefore, we focused on developing a combined model based on data decomposition.
The focus of this study was to combine the IMF components decomposed by the CEEMDAN algorithm, and to predict all IMF components separately through four different timeseries machine learning models: BP model, ARIMA model, LSTM model, and SVM model. Different characteristics were used to adapt to different models, and then the optimal prediction results corresponding to each IMF component were superimposed and summed to restore the optimal solution, so as to obtain the combination model based on the above models that was most suitable for Hangzhou PM2.5 data. Our innovation priorities were as follows:
(1)The CEEMDAN algorithm was introduced for the long-term prediction of PM2.5 timeseries data.
(2)After the IMF component was obtained using the CEEMDAN algorithm, an adaptive model was further established according to the characteristics of the IMF component, so as to improve the accuracy of long-term prediction.
(3)This method of further predictive analysis of IMF components can become a new framework, which can be applied for the prediction of more PM2.5 data, thus obtaining accurate long-term prediction results.
Modifications in the original text are as follows (page 3, line 21 - 43):
The air quality data used in this paper came from the air quality historical data query website (https://www.aqistudy.cn/historydata/), and the records of the daily average PM2.5 concentration data in Hangzhou City, Zhejiang Province were selected from December 2013 to December 2021. The sample size was 2952. The sampling interval was 1 day (the daily average data of each day were obtained by taking the arithmetic mean of the 24 h effective hourly concentration data of all single stations in Hangzhou). The PM2.5 unit was ug/m3. The first 90% of the sample data were selected for simulation training, and the last 10% of the sample data were used to test the prediction results, i.e., we used the PM2.5 concentration data for 2952 days (December 2013–February 2021) to predict the trend of PM2.5 concentration data in the next 295 days (March 2021–December 2021). Lastly, in the 295 day forecast results, we intercepted the 50 day model forecast results with more obvious comparisons for graphical display and comparative analysis. The timeseries data of PM2.5 concentration in Hangzhou are shown in Figure 1, which shows that the PM2.5 concentration in Hangzhou had obvious changes and fluctuations. The first stage of the data reveals a high concentration of PM2.5 at the end of 2013. At this time, not only in Hangzhou, but across China, the air quality was very poor, because our country did not pay attention to the problem of air pollution control before 2013. Since then, after several years of governance, the air quality has gradually improved. We chose not to delete these abnormal data from training in order to better cope with extreme data in subsequent predictions, because the actual situation is often unpredictable.
- In a 12-pages manuscript, the results only cover 3 pages from Page 8 to 10. There is no any analysis and conclusion based on the experiment results, for example, the different features of the different original algorithms, and the relation between the selected algorithms and the run-time mechanism of PM2.5 model in Hangzhou.
Response:
Sorry for our rough results and analysis part, we will rewrite this part (page 11 line 6 -page 14 line 4) to more clearly explain the prediction ability of the model after introducing the fully adaptive noise ensemble empirical modal decomposition (CEEMDAN) algorithm The improvement and the prediction performance improvement of the intrinsic mode function (IMF) generated by CEEMDAN combined with different machine learning time series models, and the reference of some important model parameter settings is added in the model introduction link (page 10 line 6 - 19), so that readers can Better to replicate the experiment, after rewriting the introduction section (page 2 line 22 -page 3 line 19), we also added a discussion on the relationship with the CEEMDAN algorithm and machine learning model and PM2.5 sample data , the CEEMDAN algorithm can completely decompose the originally highly volatile PM2.5 sample data into several lower-frequency IMF components. We use RMSE, MAE and R² as error indicators (R² is added after modification), different Which of the machine learning models combined with IMF has the better error, we consider it more suitable for the IMF of this frequency. We currently select four more mature machine learning models, and superimpose each of the most matching IMF prediction results to calculate And to get the final prediction results, we found that the accuracy of the final prediction results has increased significantly compared to before, and we think this method can be generalized.
Modifications in the original text are as follows (page 2 line 22 -page 3 line 19):
Machine learning models are generally based on basic algorithms and historical data to build predictive models that can adaptively learn model parameters, obtain laws and relationships between complex data, and conduct simulation training through part of the data to obtain models to predict future development trends. Basic parameter identification algorithms include iterative algorithms [12], particle-based algorithms [13] and recursive algorithms [14].
Machine learning models are divided into shallow networks and deep networks [15,16]. Shallow networks are short-term prediction methods based on the backpropagation (BP) neural network proposed by Ni et al. [17], which has been used for PM2.5 concentration data, daily average temperature, and other pollutant concentration data.
However, due to the simple structure of the shallow network, it can only achieve short-term prediction performance, and long-term accurate prediction results must be captured using deep neural networks.
Deep machine learning networks show strong learning ability in complex timeseries. They have good performance in capturing the high nonlinearity of timeseries data. They can highly abstract data analysis through multi-nonlinear transformation of complex structures. Timeseries problem-solving functions include long short-term memory (LSTM) [18,19], differential integrated moving average autoregressive model (ARIMA) [20,21], and support vector machine (SVM) [22,23].
Although deep machine learning models have the ability to extract accurate information in complex environments, PM2.5 concentration sequences are datasets with strong randomness and nonlinearity, and the accuracy of long-term prediction needs further development. In recent years, combinatorial approaches to data decomposition have been shown to be effective ways to improve forecasting performance, and various hybrid models have been introduced to forecast nonlinear timeseries data.
For example, the seasonal decomposition program based on loess [24,25] can perform short-term prediction of air pollen according to the decomposed seasonal components, and wavelet decomposition [26, 27] can obtain decomposition information by setting the wavelet basis function for prediction.
Fully adaptive noise ensemble empirical modal decomposition (CEEMDAN) algorithm[28,29] can decompose timeseries data into intrinsic mode function (IMF) components with different frequency characteristics [30], and sort them from high frequency to low frequency, which can greatly reduce the complexity of the original data. We believe that the predictive power of a single model is ultimately limited; therefore, we focused on developing a combined model based on data decomposition.
The focus of this study was to combine the IMF components decomposed by the CEEMDAN algorithm, and to predict all IMF components separately through four different timeseries machine learning models: BP model, ARIMA model, LSTM model, and SVM model. Different characteristics were used to adapt to different models, and then the optimal prediction results corresponding to each IMF component were superimposed and summed to restore the optimal solution, so as to obtain the combination model based on the above models that was most suitable for Hangzhou PM2.5 data. Our innovation priorities were as follows:
(1)The CEEMDAN algorithm was introduced for the long-term prediction of PM2.5 timeseries data.
(2)After the IMF component was obtained using the CEEMDAN algorithm, an adaptive model was further established according to the characteristics of the IMF component, so as to improve the accuracy of long-term prediction.
(3)This method of further predictive analysis of IMF components can become a new framework, which can be applied for the prediction of more PM2.5 data, thus obtaining accurate long-term prediction results.
Modifications in the original text are as follows (page 10 line 6 - 19):
2.1.6. Reference for Setting Important Parameters of the Model
In this model, the input dataset was the original PM2.5 timeseries data, the training set is the first 90% of the original data, and the prediction set is the last 10%, of which 10% of the training set data is selected for cross-validation of model testing, and the LSTM model was implemented using the keras package in Python 3.6. We set the most important parameter reference values to include a hidden layer with 300 dimensions, the ‘relu’ activation function, look back of 6, ‘Adam’ optimizer, batch size of 64, and 300 epochs. For the ARIMA model, we chose to use the Akaike information criterion to formulate the reasonable order of p and q, only limiting the parameter p. The maximum value of q was 3. The model automatically determined the best order within three operations. The reference order of the BP neural network model function prediction order was 30, and the hidden layer reference was three layers. In the model prediction of SVM, KernelFunction was set to ‘polynomial’, KernelScale was set to 10, and BoxConstraint was set to 5.
Modifications in the original text are as follows (page 11 line 6 -page 14 line 4):
- Results and Analysis
In this study, we first compared the prediction effectiveness of individual models for PM2.5.
The PM2.5 prediction results of the four single models are shown in Figure 8, and the error results are shown in Table 1. It can be concluded that the basic trend of the future PM2.5 concentration could be predicted simply by using single models, but there were clearly large errors, especially at the peaks and troughs. The ARIMA model was the best model for predicting PM2.5 in Hangzhou among the four single models, with an RMSE of 10.09, an MAE of 7.51, and an R2 of 0.46.
Figure 8. Comparison of single model prediction results.
Table 1. Single model error comparison.
|
|
LSTM |
ARIMA |
BP |
SVM |
|
RMSE |
10.27 |
10.09 |
10.55 |
10.61 |
|
MAE |
7.88 |
7.51 |
8.21 |
7.93 |
|
R2 |
0.42 |
0.46 |
0.37 |
0.43 |
We then optimized the single models using the CEEMDAN algorithm due to its predictive power. Figure 9 and Table 2 show the prediction and error comparison results for the CEEMDAN–LSTM, CEEMDAN–ARIMA, CEEMDAN–BP, and CEEMDAN–SVM models.
Figure 9. Comparison of prediction results using coupled models based on CEEMDAN algorithm.
Table 2. Comparison of coupled model errors based on CEEMDAN algorithm.
|
|
CEEMDAN–LSTM |
CEEMDAN–ARIMA |
CEEMDAN–BP |
CEEMDAN–SVM |
|
RMSE |
7.28 |
6.27 |
6.90 |
6.27 |
|
MAE |
5.71 |
4.95 |
5.48 |
4.88 |
|
R2 |
0.70 |
0.74 |
0.67 |
0.77 |
Without the CEEMDAN algorithm, the predictive ability of the ARIMA model was better than that of the SVM model, whereas, with the CEEMDAN algorithm, the prediction ability of the CEEMDAN–SVM model was better than that of the CEEMDAN–ARIMA model. Their RMSE values were the same, but the MAE values of the CEEMDAN–SVM model were better. Therefore, we chose to compare the prediction results of each modal component after decomposition using the CEEMDAN algorithm in detail.
Table 3. Comparison of RMSE values of split result graphs based on CEEMDAN algorithm.
|
|
LSTM (RMSE) |
ARIMA (RMSE) |
SVM (RMSE) |
BP (RMSE) |
|
IMF1 |
5.84 |
6.16 |
6.12 |
6.90 |
|
IMF2 |
1.76 |
1.53 |
1.80 |
1.52 |
|
IMF3 |
0.27 |
0.25 |
0.71 |
0.29 |
|
IMF4 |
0.15 |
0.04 |
0.43 |
0.05 |
|
IMF5 |
0.10 |
1.57× 10−3 |
0.20 |
1.9 × 10−3 |
|
IMF6 |
0.04 |
6.66 × 10−5 |
0.46 |
1.1 × 10−3 |
|
IMF7 |
0.06 |
3.72 × 10−6 |
0.77 |
1.2 × 10−3 |
|
IMF8 |
0.01 |
3.81 × 10−8 |
0.14 |
3.21 × 10−4 |
|
IMF9 |
0.13 |
1.38 × 10−8 |
1.34 |
0.09 |
|
Residual |
0.00 |
0 |
0 |
0 |
Table 4. Comparison of MAE values of split result graphs based on CEEMDAN algorithm.
|
|
LSTM (MAE) |
ARIMA (MAE) |
SVM (MAE) |
BP (MAE) |
|
IMF1 |
4.70 |
4.97 |
4.75 |
5.51 |
|
IMF2 |
1.37 |
1.20 |
1.31 |
1.17 |
|
IMF3 |
0.19 |
0.17 |
0.53 |
0.21 |
|
IMF4 |
0.13 |
0.02 |
0.35 |
0.03 |
|
IMF5 |
0.05 |
1 × 10−3 |
0.14 |
1 × 10−3 |
|
IMF6 |
0.03 |
2.17 × 10−5 |
0.38 |
7.83 × 10−4 |
|
IMF7 |
0.05 |
7.36 × 10−7 |
0.68 |
8.78 × 10−4 |
|
IMF8 |
0.01 |
7.88 × 10−9 |
0.10 |
3.10 × 10−4 |
|
IMF9 |
0.09 |
9.35 × 10−9 |
1.34 |
0.074 |
|
Residual |
0.00 |
0.00 |
0.00 |
0.00 |
Table 5. Comparison of R2 values of split result graphs based on CEEMDAN algorithm.
|
|
LSTM (R2) |
ARIMA (R2) |
SVM (R2) |
BP (R2) |
|
IMF1 |
0.20 |
0.16 |
0.17 |
0 |
|
IMF2 |
0.88 |
0.91 |
0.88 |
0.91 |
|
IMF3 |
0.99 |
1.00 |
0.98 |
1.00 |
|
IMF4 |
1.00 |
1.00 |
0.99 |
1.00 |
|
IMF5 |
1.00 |
1.00 |
1.00 |
1.00 |
|
IMF6 |
1.00 |
1.00 |
1.00 |
1.00 |
|
IMF7 |
1.00 |
1.00 |
1.00 |
1.00 |
|
IMF8 |
1.00 |
1.00 |
1.00 |
1.00 |
|
IMF9 |
1.00 |
1.00 |
1.00 |
1.00 |
|
Residual |
0.00 |
0.00 |
0.00 |
0.00 |
We used the CEEMDAN algorithm to completely decompose the original PM2.5 concentration data in Hangzhou into nine IMF components and residuals according to their frequency from high to low, and then substituted each IMF component and residual into the LSTM, ARIMA, SVM, and BP neural network models for prediction. The final prediction result was obtained by stacking and summing. The comparison results are shown in Table 3, Table 4, and Table 5. The LSTM network performed best in IMF1 component prediction. The BP neural network performed best in predicting the IMF2 components. The prediction performance of the ARIMA model was excellent for the IMF3–9 components. Since the residual value was miniscule, the value of all models when predicting the residual was infinitely close to 0. It can be concluded that the model corresponding to each data type with the best performance was different, and no model could be used for all components. Therefore, data types with different frequencies should be matched with different models for prediction, and the optimal solution for each IMF component prediction was obtained by constructing CEEMDAN–LSTM–BP–ARIMA model. The prediction results obtained from the experiment are shown in Figure 10, and the error results are shown in Table 6. The fitting degree in the figure is high, and the various error indicators are significantly better than all the previous models. It can be concluded that the CEEMDAN–LSTM–BP–ARIMA model is an ideal model for predicting PM2.5 concentration in Hangzhou.
Figure 10. CEEMDAN–LSTM–BP–ARIMA prediction comparison.
Table 6. CEEMDAN–LSTM–BP–ARIMA model error comparison.
|
|
CEEMDAN–LSTM–BP–ARIMA |
|
RMSE |
5.9 |
|
MAE |
4.63 |
|
R2 |
0.79 |
- After reviewing this manuscript, we still don’t know the reason why authors choose this combination of the algorithms, when there is only an exhibition of the experiment accuracy. Because the good accuracy of your method must be explainable and adaptable, which need to be proved by lots of experiment and analysis.
Response:
Sorry again for the inconvenience, we have rewritten the first part about the CEEMDAN algorithm, hoping to express it more clearly so that you can understand the advantages of this combination of algorithms (page 4 line 6 - 18), and what we want to express is the improvement of prediction accuracy brought by the CEEMDAN decomposition algorithm to the machine learning time series prediction model, a deep learning network can theoretically model any complex nonlinear time series data, but due to the limitation of the amount of training data and network scale, It cannot obtain enough accuracy to predict PM2.5 data, but in the case of the same amount of PM2.5 data, combining the CEEMDAN algorithm with the four time series machine learning models in this paper can greatly improve the prediction accuracy of the model rate, and we don’t think it’s an accident, it’s a new approach that can generalize, and we believe that more time series machine learning models can use this approach to improve forecast accuracy. The CEEMDAN algorithm combines the decomposition components of the original PM2.5 data with different machine learning models, which will also bring us new improvements. Which can provide a new ideal method for PM2.5 prediction, and hope to achieve greater breakthroughs in the future. Including models with higher prediction accuracy has the potential to achieve further effects through the CEEMDAN algorithm, or to apply it to a wider range of data types.Finally, including the conclusion part (page), we have also made corresponding changes, hoping to bring you a better reading experience.
Modifications in the original text are as follows (page 4 line 6 -18):
The CEEMDAN algorithm is mostly used in the field of forecasting. It can completely decompose the original data with strong volatility into several intrinsic mode function (IMF) components with different frequency characteristics, thereby reducing the volatility of the data and improving the prediction accuracy. The CEEMDAN algorithm has three major advantages. The first is its completeness. In other words, the original data can be obtained by adding and summing the components decomposed by the algorithm, which is beyond the reach of many decomposition algorithms. Secondly, the CEEMDAN algorithm has a faster calculation speed, which can effectively improve the operation speed of the program. Lastly, the CEEMDAN algorithm has a better modal decomposition effect, preventing the occurrence of multiple low-frequency components with small amplitudes, which are of little significance for data analysis.
Modifications in the original text are as follows (page 14 line 5 -30):
In recent years, air quality problems have had a serious impact on people’s normal life. Environmental problems such as PM2.5 have received more and more attention, and PM2.5 is characterized by strong multilateral and strong randomness. Thus, accurate long-term PM2.5 concentration prediction remains a formidable challenge for us.
In this study, we proposed a way to combine the CEEMDAN algorithm with the LSTM model, ARIMA model, BP neural network, and SVM model to predict the PM2.5 concentration in Hangzhou. The results of various evaluation indicators showed that all models based on the CEEMDAN algorithm improved the prediction accuracy to varying degrees compared with the original simple models. The introduction of the CEEMDAN algorithm can provide new inspiration for PM2.5 prediction, and the CEEMDAN algorithm can perhaps be combined with additional timeseries machine learning models. In this experiment, the predictive performance of the coupled model was higher than that of the single model.
Secondly, we discovered a new application of the IMF components obtained using the CEEMDAN algorithm. We carried out LSTM, ARIMA, BP neural network, and SVM modeling and prediction for each IMF component of the Hangzhou PM2.5 concentration timeseries data. The optimal prediction results of the components were added and summed. The CEEMDAN–LSTM–BP–ARIMA model obtained the best prediction results of PM2.5 concentration in Hangzhou in this paper. Compared with the other models, the long-term prediction accuracy was significantly improved. Applying a single model is not optimal; we found that the best models differed according to IMF components, whereby a combination of timeseries machine learning models obtained the best prediction accuracy. We believe that this method also has good generalizability and can be used to predict additional characteristics such as wind speed and other pollutant concentrations.
We appreciate your warm work earnestly and hope that the corrections will meet with approval.Once agan, thank you very much for your comments and suggestions.
Yours sincerely:
WenChao Ban
Corresponding author
Name:LiangDuo Shen
Address: No. 1 Haida South Road, Hangzhou City, Zhejiang Province, China
E-mail:slduo@163.com
Reviewer 2 Report
General:
The manuscript describes a scheme in which four known time series prediction algorithms are applied to components of a daily PM2.5 concentration series, and the predictions of the components are then superimposed to give a final prediction of the daily PM2.5 series. The author show that applying the CEEMDAN decomposition pre-processing significantly improves the prediction compared to predictions by the same algorithms applied to the original series. The best result is achieved when the superposition is of predictions produced by applying the best algorithm for each component. The basic idea is nice as a methodology for predicting time series using only its past values. However, the worthy motivation provided by the authors is specified as "Therefore, it is necessary to establish accurate, reliable, and effective models to make predictions of atmospheric pollutant concentrations in a certain period of time in the future." My main reservation with this work is that most probably regression models, using additional variables, can perform better than the model found best by the authors, but using only the data of the time series itself in the prediction. A second problem with the authors' well-defined goal is that they never mention what is this "certain period of time in the future" to which they produce their predictions. It is most important in a paper describing a new prediction model to mention the prediction time and to compare how the prediction performance falls as the prediction is made for time points further in the future. In addition, the paper is not well written. It is not following the guidelines as specified by the instruction for authors, the explanations are in many instances not clear and the English writing style could be improved.
Major comments:
The prediction time must be mentioned. It will be a good idea to consider a few prediction times (e.g., 1,2, 5 days ahead), compare the results and discuss them.
In the conclusions, the authors say "In this paper, the CEEMDAN-LSTM-BP-ARIMA model was an ideal method to predict PM2.5 in Hangzhou city." This is true, but is it the best possible model to predict PM2.5 in Hangzhou city? How regression models, which use additional input variable, perform compared to the CEEMDAN-LSTM-BP-ARIMA model? Given the motivation for the work as specified by the authors, the best predictions provided by the methodology described in the manuscript must be compared to a method using additional input variables. A good option would be using the back propagation with input including meteorological variables (e.g., wind direction components, atmospheric stability, mixing layer depth etc., which should be available in a city the size of Hangzhou) and if possible, also traffic volumes (cars, trucks, trains and may be shipping in the Grand Canal). Only if the manuscript's best prediction method is found superior, the motivation for the work (as specified by the authors) is justified. An alternative is to present the work as a technical study introducing a prediction method and avoiding the environmental motivation.
The writing of the manuscript must be improved. The aid of a native English speaker should be sought to improve the English writing style, and the explanations in the methods and results sections must be made more clear. Please note that the publications requires in the instructions for authors regarding the methods is that "They should be described with sufficient detail to allow others to replicate and build on published results." This is clearly not the case with this manuscript. Neither a mention of where the data can be obtained nor sufficient detailed explanations are given so that readers can try and replicate the work.
Minor comments:
Why the authors selected the EMD algorithm and not any other decomposition? The answer to this question might have been given in the first paragraph of section 2.1.1, but this paragraph is not clear at all. It should be re-written.
The instructions for authors clearly say, "Acronyms/Abbreviations/Initialisms should be defined the first time they appear in each of three sections: the abstract; the main text; the first figure or table. When defined for the first time, the acronym/abbreviation/initialism should be added in parentheses after the written-out form." The authors ignored this good suggestion and as a result, reading the manuscript, which is full of acronyms, was very difficult.
It would have been much easier to provide comments had line numbers were added to the manuscript.
"... which is based on the cornerstone of RNN and ..." Not clear what is RNN. This acronym has not been defined previously in the manuscript, nor a comprehensive list of acronyms provided anywhere in it. Given the many acronyms used, it would be a good idea to provide such a list (beyond it being a requirement specified in the instructions to authors).
"... decomposes the signal into a series of IMF components and residuals.." Not clear what is IMF. I guess it means "intrinsic mode functions" but cannot be sure. The authors should spell out each acronym and provide references to any process they carry out which the readers might not be familiar with.
"Finally, the model prediction results of 50 days were randomly intercepted." Not clear what "intercepted" mean in this context. The sentence is not clear.
The authors used the last 10% of the data points for testing. More comprehensive testing would be using 10 cross-validations in which, in turn a 10% section is left out, it is predicted by a model (trained on the rest 90%), and the results are then merged for the model performance calculated over the full study period. Given that the model performance measures by the different models are quite close, such more comprehensive testing is a good idea.
Figure 1: Can the authors explain the very high PM2.5 concentrations at the very beginning of the record? Shouldn't these data be removed from the training, given how different are they from the rest?
An area map should be provided. It should show the location of the observation station and the possible local sources in the surrounding area (roads, water-ways, nearby industrial plants emitting PM, etc.).
The author used daily data. Is this time-scale the original measurement interval? If the observations are carried out on a finer scale (I guess it is hourly, as is usually the case in China), how many missing data are in the original series and how they are distributed? Missing data which are not randomly distributed may have an impact on the modelling.
Why the choice of q = 1.2? How the decomposition change if different values of q are selected?
Sections 2.1.2, 2.1.3, 2.1.4 and 2.1.5 provide very brief descriptions of the four forecasting methods which are used. Clearly there is no need for detailed descriptions given their long history and the vast literature in which they are described. However, I would expect (a) citations of the relevant literature which fully describe each method and (b) setting the explanation in the context of the current work. For example, what are the X1, X2 and X3 in the input to the back propagation neural net in this work? And what are Y1 and Y2? How the fit level in the training stage was determined? Similar questions can be raised by readers regarding each of the methods, and answers should be provided (possible in a supplementary material section). Dumping on the reader a general description of a method does not promote understanding of how it is actually used in the study.
The two model evaluation measures (section 2.1.6) of the prediction performance are similar in the sense that they both measure the size of the mean prediction error (one linear and one quadratic). Not surprisingly, the order of the model performance in both Tables 1 and 2 is similar using RMSE or MAE. Maybe it would be better to provide two measures (or add a third) which with a different. For example, commonly the correlation coefficient or the R^2 are used for this purpose.
Figure 6 and 7: What is the prediction time? That is, how far in future the models forecast a PM2.5 concentration value?
Figure 7: The predictions here do not seem to be superior to those in Figure 6. They might have lower RMSE and MAE, but timing and intensity of the peaks (which are the more important to predict correctly) are not reproduced as well as in Figure 6. Please comment.
Table 1: What are the units of the RMSE and MAE model performance measures? I guess they are in μgm^(-3) units, but this should be clearly said in the table caption.
".. while the RMSE value of the four coupled models based on the CEEMDAN algorithm in Table 2 is
10.09." The sentence is not clear. It is probably wrong (the value 10.09 appears in Table 1, not Table 2).
Figures 8 and 9: I think that these figures and the related description in the text should appear in the methods section.
I assume that the CEEMDAN-LSTM-BP-ARIMA uses for each IMF the prediction model that best predicts it, and then superimposes the results. If indeed this is what the authors did, it is a nice and good idea. However, the explanation in the text does not make clear that this is indeed what they have done.
Table 5: First, the results (2 values) of this table should be incorporated in Table
1 and/or two. That way, it will be easy to compare the improved performance to the ones presented before. Second, the table heading says CEEMDAN-LSTM, which is not correct and does not agree with what the caption says.
The first part of the conclusions section is a summary of previous results, not conclusions.
The first part of the conclusions mention an increase in the RMSE and MAE of the modle using CEEMDAN decomposition. The RMASE and MAE actually decreased (the smaller the RMSE and MAE the better).
Author Response
please check the attached response.

Round 2
Reviewer 1 Report
first,the authors work hard to improve the manuscript. but, I think that there is long distance for publishing the muanuscript. there are several reasons: 1. why did you choose the combination of the models (e.g.,CEEMDAN and LSTM ). the only reason is that the results from CEEMDAN–LSTM–BP–ARIMA is optimal from your manuscript, which is unreliable while experiment being carried out in a so small and local area. 2. following the aforementioned reason, you need discuss the reason why CEEMDAN–LSTM–BP–ARIMA is optimal; regettably, you just detailed the resluts from the different models without the profound disscusion (for example, the structuce of models is good based on the transmission mechanism of air pollution). at last, 3 no enough large experiment, no disscusion based on the physical mechanism result in the unreliable applicablity of the proposed method.
considering the above unanswered questions. i don't agree with the pubilcation of the manuscript in current state.
Author Response
Response to Reviewer 1 Comments
Point 1:why did you choose the combination of the models (e.g.,CEEMDAN and LSTM ). the only reason is that the results from CEEMDAN–LSTM–BP–ARIMA is optimal from your manuscript, which is unreliable while experiment being carried out in a so small and local area.
Point 3:no enough large experiment, no disscusion based on the physical mechanism result in the unreliable applicablity of the proposed method.
Responses 1 and 3: The CEEMDAN-LSTM-BP-ARIMA results are indeed the best in my manuscript, but this does not mean that the laboratory is unreliable. In recent years, many studies have shown that deep learning methods are reliable for the application of air pollution, and the combination of a single machine learning model and data decomposition has been proved to be an effective method to achieve prediction. Deep cnn-lstm model for Smart City particulate matter (PM2.5) prediction. Sensors 2018, 18, 2220.; Rojo; Rivero; Romero-Mott; Fernandez-Gonzalez; Perez-Badia, R. Modeling pollen time series using a loess smoothing based seasonal trend decomposition procedure. International J. Biometalyl Alcohol.2017, 61, 335 -- 348. Xiong Tao, Li Ke, Seasonal forecasting of agricultural prices based on hybrid STL and ELM methods: Evidence from the Chinese vegetable market. Neural Computing 2018, 275, 2831 -- 2844. Cheng Yong; Zhang He; Liu Zhi; Chen Lijun; A hybrid algorithm for short-term PM2.5 prediction in China. Ambient Environment, 2019, 200, 264 -- 279. 5. The hybrid model based on wavelet transform and the improved deep learning algorithm are used. IEEE Access 2019, 7, 142814 -- 142825] has also introduced various hybrid models to predict nonlinear time series data. What I do is to innovate on the basis of previous models and some new ideas, such as the introduction of CEEMDAN algorithm, Different machine learning models are used to start further analysis of IMF components, etc. Our experimental data were also conducted for a full 8 years, and the prediction set reached 295 days, including the cross-validation of the model training data between the two. We believe that the conclusion obtained by this method is reliable.
Point 2: According to the above reasons, you need to discuss the reasons why CEIBS Dan -LSTM-BP- has the best horse; In the reprint, you just describe in detail the refutations from the different models without deep discussion (for example, the structure of the model is good based on the transmission mechanism of air pollution). The last
Response 2: Thank you very much for your comments, but we are based on the experiment on the prediction of pollutant concentration, the purpose is to put forward a more accurate prediction results to community service, and we discussed the reason, the optimal model to adapt to different frequency wave data are used to different machine learning model, class communication mechanism we might think that there is no need for further discussion, This may not necessarily be relevant to what we are writing.
Reviewer 2 Report
I carefully read the revised version and the authors' responses to the review. Clearly the authors took care to improve technical aspects of the presented work. The language is indeed much better, acronyms defined, and line number can facilitate an easier review report. Longer descriptions of the ML methods were given and an additional model performance measure was considered. However, I do not feel that there was any change in the work in essence. Many words were added to the manuscript, especially in the description of the machine learning prediction methods, but the main issue is that I, as an air pollution modeller, cannot see a motivation to consider using the work as long as not clear to me what it actually does and how this compared to other benchmark methods. I clearly understand the point the authors make: the combined process of starting with CEEMDAN pre-processing using the best ML method for the prediction of each IMF and then superimposing the results is better, than using the combined process using each method consistently for all the IMFs, and much better than not using CEEMDAN at all. But I still did not get answers to my simple questions in the original review: (1) what is the forecast lead time? and (2) how forecasting by the proposed methods is compared to other prediction methods? I will make clear the two questions below as it seems there is misunderstanding about them with the authors. As a general opinion, I reiterate what I said in the first review, the work presents a nice idea. The motivation for the work though, as presented by the authors, is its potential for practical PM2.5 concentration prediction. Not understanding the forecast lead time I cannot recommend this method for practical use. Knowing the forecast lead time, we need to know the proposed method is at least as good in such prediction as the most simple benchmarks.
I will make clear the two questions. The forecast lead time is the time lag between the time of the last observed data used to train the model and the time of the prediction. So if Xi in equation 7 is a PM2.5 value at day i, how many days forward in time will the algorithm predict, i+1, i+2, i+3 ,etc? For example, if I have a PM2.5 time series up to 31-12-2022, can the method predicts to 01-01-2023, 02-01-2023, 03-01,2023, etc.? To what lead time the results in Table 1-5 refer? Prediction from 31-12-2022 to 01-01-2023, for example, is a one day lead time forecast.
When the forecast time issue is made clear then it's time to compare the performance of the proposed model to the performance of other models currently in use. At the least, I would expect a comparison to the two most basic benchmark methods (a) consistency. That is, setting the prediction of PM2.5 concentration tomorrow to be the value today. (b) Seasonality. Setting the value at each day to that of the mean of the values in the same day of the year in previous years (that is, PM2.5 concentrations in 01-01-2023 be the mean values of the PM2.5 concentration in 01-01-2013, 01-01-2014, 01-01-2015, etc.). A much more challenging and meaningful comparison would be to the simplex (Sugihara and May 1990) or the Smap methods (Sugihara
1994) which are based on the empirical dynamic systems approach. Both methods were implemented in the freely available R package rEDM. If the out of sample performance of the method proposed by the authors cannot beat at least the two most basic methods above, what's the point in publishing it?
There are other issues, few of them will be mentioned below, but lack of answers to the two basic question above is what prevents me from recommend acceptance of the work for publication is Sustainability.
Minor issues:
The authors trained the models on 2657 data points and tested them on 295 out of sample data points. That is fine although my recommendation was of a 1-fold cross-testing. This process is much more comprehensive and less biased than arbitrarily selecting the last 10% of data for testing. To make sure what 10-fold cross-testing means, it is using the data points 1-2657 for training and data points 2658-2952 for testing, then use data ponts 1-2362 and 2657-2952 for training and data points 2363-2656 for testing, then use data points 1:2067 and data points 2363-2952 for training and data points 2068-2362 for testing, and so on for each 10% testing section. At the end the testing predictions are added and compared to the observed data.
The descriptions of the ML methods are now longer but are not better explaining their usage in the context of the work. They feel more like sections cut from some sources and pasted into this work with no attempt to set the description in its context. I think, and it is also a requirement of Sustainability, that these descriptions should be re-written so that a reader can exactly repeat the work (i.e, full explanations how each method was implemented and how the many parameters were selected). These description do not have to be presented in the main work. They can be given in a complementary material.
References:
Sugihara, G., May, R., 1990. Nonlinear forecasting as a way of distinguishing chaos from measurement error in data series. 344, 734-741.
Sugihara, G., 1994. Nonlinear forecasting for the classification of natural time series. Phil. Trans. R. Soc. Lond. A 348, 477-495.
~
Author Response
Response to Reviewer 2 Comments
Point 1: what is the forecast lead time?
Response 1 : The forecast lead time is 1 day, and the results in Tables 1-5 show the error indicators for the entire forecast period (295 days).
Point 2:how forecasting by the proposed methods is compared to other prediction methods?
Response 2 : Thank you very much for the reviewer's comments, but maybe we don't agree with the way you described the comparison, is this approach too old-fashioned,many studies in recent years have shown that deep learning methods are reliable for applications in the field of atmospheric pollution, and both single machine learning models and combined methods of data decomposition have been shown to be an effective way to achieve predictions.[Huang, C.J.; Kuo, P.H. A deep cnn-lstm model for particulate matter (PM2.5) forecasting in smart cities. Sensors 2018, 18, 2220.;Rojo, J.; Rivero, R.; Romero-Morte, J.; Fernandez-González, F.; Perez-Badia, R. Modeling pollen time series using seasonal-trend decomposition procedure based on LOESS smoothing. Int. J. Biometeorol. 2017, 61, 335–348.;Xiong, T.; Li, C.; Bao, Y. Seasonal forecasting of agricultural commodity price using a hybrid STL and ELM method: Evidence from the vegetable market in China. Neurocomputing 2018, 275, 2831–2844.;Cheng, Y.; Zhang, H.; Liu, Z.; Chen, L.; Wang, P. Hybrid algorithm for short-term forecasting of PM2.5 in China. Atmos. Environ. 2019, 200, 264–279. ;Qiao, W.; Tian, W.; Tian, Y.; Yang, Q.; Wang, Y.; Zhang, J. The forecasting of PM2. 5 using a hybrid model based on wavelet transform and an improved deep learning algorithm. IEEE Access 2019, 7, 142814–142825],we believe that a comparative analysis with machine learning methods that are already mature enough to be used is sufficient.
Point 3:The authors trained the models on 2657 data points and tested them on 295 out of sample data points. That is fine although my recommendation was of a 1-fold cross-testing. This process is much more comprehensive and less biased than arbitrarily selecting the last 10% of data for testing. To make sure what 10-fold cross-testing means, it is using the data points 1-2657 for training and data points 2658-2952 for testing, then use data ponts 1-2362 and 2657-2952 for training and data points 2363-2656 for testing, then use data points 1:2067 and data points 2363-2952 for training and data points 2068-2362 for testing, and so on for each 10% testing section. At the end the testing predictions are added and compared to the observed data.
Point 4: The description of ML methods is now longer, but does not better explain their use in a working context. They feel more like sections cut and pasted into this work from some source, without trying to set the description in its context. It is also, I believe, a requirement of sustainability that these descriptions should be rewritten so that the reader can repeat the work exactly (i.e., a full explanation of how each method was implemented and how many parameters were chosen). These descriptions do not have to be presented in the main work. They can be given in the Supplementary material.
Response 4: We rewrote the methods section and parameter setting section to optimize the research background, including the description of the application of existing mature machine learning techniques to air pollution research, and chose to mark the changes in the original text in red.